# Activated Corrosion Products Evaluations for Occupational Dose Mitigation in Nuclear Fusion Facilities

**Nicholas Terranova** [1,*] **, Simona Breidokaité** [2] **, Gian Marco Contessa** [1] **, Luigi Di Pace** [3] **, Claudia Gasparrini** [4,5] **, Tadas Kaliatka** [2] **and Giovanni Mariano** [1]

1    ENEA Fusion and Technology for Nuclear Safety and Security Department, CR Frascati, Via Enrico Fermi 45, I-00044 Frascati, Italy; gianmarco.contessa@enea.it (G.M.C.); giovanni.mariano@enea.it (G.M.)
2    LEI, Laboratory of Nuclear Installation Safety, Lithuanian Energy Institute, Breslaujos Str. 3, LT-44403 Kaunas, Lithuania; simona.breidokaite@lei.lt (S.B.); tadas.kaliatka@lei.lt (T.K.)
3    RINA Consulting—Centro Sviluppo Materiali S.p.A. Roma, Via di Castel Romano 100, I-00128 Rome, Italy; luigi.dipace@gmail.com
4    Consorzio RFX, CNR, ENEA, INFN, Università di Padova, Acciaierie Venete S.p.A., I-35127 Padova, Italy; claudia.gasparrini@igi.cnr.it or c.gasparrini14@imperial.ac.uk
5    Department of Materials & Centre for Nuclear Engineering, Imperial College London, London SW7 2AZ, UK
*    Correspondence: nicholas.terranova@enea.it

**Abstract:** Activated corrosion products generation in primary heat transfer systems of nuclear fusion facilities is a relevant radiological source term for occupation dose assessments. The formation of the Chalk River Undefined Deposit, already well known in nuclear fission power plants, represents a significant safety issue in fusion applications due to the intense high energy neutron fluences (about 14 MeV in Deuterium-Tritium operation). The activated corrosion products formation is a multi-physical problem. The combined synergy of activation, corrosion, dissolution, erosion, ejection, precipitation, and transport phenomena induces the contamination of coolant loop regions located outside the bio-shield, where scheduled worker operation might take place. The following manuscript shows how activated corrosion products are evaluated for the nuclear fusion power plant design under investigation by the Safety and Environment Work Package (WPSAE) of the Eurofusion Consortium (i.e., the European Demonstration power plant, EU-DEMO). The major issues in activated corrosion products estimations are here exposed and the main results for mass and activity inventories are briefly shown for some main Primary Heat Transfer Systems of EU-DEMO.

**Keywords:** corrosion; ACP; DEMO; occupational dose

## 1. Introduction

The estimation of the occupational dose (ORE, Occupational Radiation Exposure) in nuclear power plants is a fundamental aspect of the whole safety evaluation of the installation. In safe-by-design projects, realistic estimates of the collective doses in terms of person·mSv/y are necessary requirements in the radiological safety optimization process. Accurate and realistic collective dose values can be achieved only if a precise determination of the preventive and corrective maintenance activities is available, together with the associated dose rates due to the radiological source terms of the system. In nuclear fusion installations as ITER [1] (International Thermonuclear Experimental Reactor) or future EU-DEMO [2] (European DEMOnstration power plant), ORE analyses are performed since the conceptual design of the machines.

The determination of the collective dose in scheduled preventive and corrective maintenance operations in experimental fusion plants is radically affected by the inventory of Activated Corrosion Products (ACP). The estimation of ACP inventory and the subsequent water chemistry optimization allows to reduce the impact on collective dose, participating actively in the ALARA process (As Low As Reasonably Achievable) since the design phase.

The determination of the ACP inventory is then a crucial aspect of the safety assessment and licensing, being an open research field in several nuclear application domains [3–11].

The CRUD (Chalk River Undefined Deposit) generation is a well-known problem in nuclear fission power plants. It assumes particular significance in nuclear fusion experimental facilities due to the higher neutron spectra which open further nuclear transmutation reaction channels in structural materials used for cooling and shielding.

The ACP formation in water cooled loops is a complex multi-physical process. Two layers are generally produced on the pipe surfaces. The inner oxide layer is a more adherent dense film, richer in chromite, which has the particular feature of being more attached to the underlying metal. The outer oxide or deposit is characterized by tetrahedral crystals loosely attached to the surface which agglomerate through precipitation processes of ions and particles in solution and suspension in the fluid, respectively. This outer oxide is characterized by a specific porosity which allows releases between the metal and the working fluid. Mechanical erosion might also take place, enhancing the neutron activated material transport.

Mobile particles, ions, and agglomerate can be activated in the under-flux region of the experimental facility and transported in the ex-vessel vaults where working personnel may have access for maintenance operations. Chemical and physical phenomena, such as dissolution or erosion, can mobilize activated materials which can be transported and form deposits in components not directly exposed to neutron radiation fields (e.g., heat exchangers, piping, valves, filters, and pumps). Such oxide deposits may contain gamma-rays emitters which must be quantified during the safety assessments involving occupational exposure. More than 90% of the occupational exposure may derive from ACP contamination in Pressurized Water Reactors (PWR) [12]. Similar conclusions might be drawn from experimental fusion facilities for which ACP estimations are under definition.

In ACP formation, multi-physical mechanisms including corrosion, release, diffusion, dissolution, precipitation, erosion, abrasion, activation, decay, transport, etc., take place. The complexity of the process requires the development and employment of dedicated software tools. In recent years, several codes such as PACTOLE [3], TRACT [13], PACTITER [14], CATE [8], and OSCAR [6] have been developed to estimate ACP in nuclear power plants cooling systems. The main goal of the present work is to present, in Section 2, how ACP can be evaluated for experimental fusion installations using OSCAR-Fusion, a code developed by the French Atomic Commission in collaboration with Electricité de France and Framatome. In preliminary safety investigations performed by the Safety and Environment Work Package (WPSAE) of the Eurofusion Consortium, the WCLL (Water Cooled Lothium Lead) concept of EU-DEMO has been identified to be particularly prone to produce ACP in its primary water cooling circuits. ACP evaluations, using dedicated calculation tools derived from fission power reactors, turned to be necessary to trigger the optimization process aimed at reducing the occupational dose due to external gamma exposure in ex-vessel vaults. The general procedure, described in the following sections, envisages the estimation of neutron activation reaction rates, the ACP contamination evaluation and, finally, the photon transport calculation for shut down dose rate maps definition. Some preliminary results and the application of the first two steps performed for the Divertor Primary Heat Transfer System (DIV-PHTS) are presented in Section 3.

## 2. Materials and Methods

### 2.1. General Modelling Approach

ACP assessments in fusion tokamak PHTS is a relevant safety issue. The general methodology consists essentially in modelling the system as simplified regions connected in a closed loop [5] (see Section 2.2). Corrosion laws and the physical processes involved must be defined, the activation reaction rates used as source terms in the Bateman equation must be calculated (see Section 2.4). The actual ACP inventory calculation can be addressed using the OSCAR-Fusion code. OSCAR-Fusion allows simulating contamination in nuclear

reactor cooling system due to activated materials which are released in the cooling water. The assessment requires five main inputs:

- Cooling loop modelling to be used as geometrical input in OSCAR-Fusion. The loops should be subdivided in multiple regions with assigned operational, chemical and radiation boundary conditions representing the actual behaviour of the system.
- Corrosion and release model input parameters to define the governing corrosion kinetics equations.
- Activation reaction rates for the OSCAR-Fusion regions under neutron flux. These should be intended as mono-energetic homogenized reaction rates which should reproduce the actual base metal and oxide activation under neutron flux in the DIV region.
- Water chemistry conditions. OSCAR-Fusion simulates the behavior of ultra-pure water, with assigned concentrations of hydrogen, oxygen, boron and lithium.
- Operational scenario in terms of water flow rate and power evolution time schedule.

Mono-energetic neutron reaction rates must be previously calculated. OSCAR-Fusion solves the Bateman equation for radioisotope nuclear decay to calculate the inventory of the radioactive materials. In fusion neutronics, reaction rates are generally calculated using Monte Carlo, more details on that will be provided in Section 2.4.

Once ACP inventories in terms of masses and specific activities are estimated, source terms for Monte Carlo photon transport calculations must be defined. In the present work, we focus on the ACP assessment, showing results on the ACP inventory which are quite significant in estimating the radiological hazard.

### 2.2. Cooling Loop Modelling

The main input parameters needed for the model definition are the following:

- Geometrical and thermal-hydraulic data;
- Material properties (including oxide morphology);
- Operational data.

OSCAR-Fusion requires the definition of 1D regions with assigned conditions of bulk temperature, wall temperature, pressure, velocity, wet surface, hydraulic diameter, and flow rate. Metal, inner oxide and outer oxide thickness, roughness, and composition must be defined according to the industrial manufacturing and the pre-commissioning tests of the machine.

In tokamaks ACP analysis, the machine is usually subdivided in ex-vacuum-vessel and in-vacuum-vessel components, which are normally under direct neutron flux. The formers belong to what is normally called Balance-of-Plant (BOP). The BOP represents the conventional part of the power plant which has several common features with fission power plants. It includes heat exchangers, pumps, valves and steam generators which are normally nodalized in multiple regions according to the temperature gradients, pressure drops and heat loads (see EU-DEMO Divertor PHTS in Figure 1 as an example). Another crucial component of the BOP is the Chemical and Volume Control System (CVCS) which gathers the water filtering and purification functions of the plant.

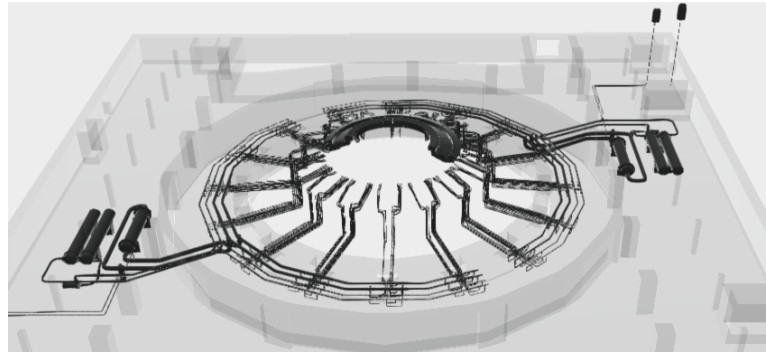

**Figure 1.** Representation of the EU-DEMO Divertor PHTS CAD model.

In tokamaks, the cooling channels under direct neutron flux could have architectures and layouts of high levels of complexity. They are generally made of arrays of cooling channels with complex manifolds and distributors made of specific nuclear grade materials (i.e., RAFMS, Reduced Activation Ferritic/Martensitic Steel, see Figure 2). Wet surfaces can be measured using CAD fluid/pipe interface assessment tools.

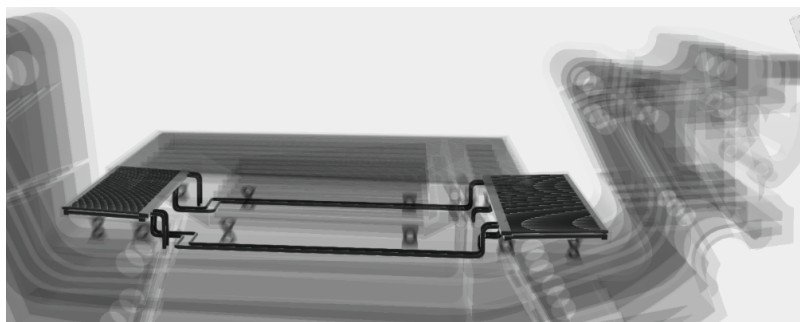

**Figure 2.** Representation of the EU-DEMO Divertor cassette CAD model. Liner and reflector plates cooling channels are highlighted.

In the EU-DEMO Divertor PHTS, the cassette body under neutron flux is made of EUROFER-97 which allows lower neutron activation. The Plasma Facing Unit channels are made of CuCrZr with pure Cu tapes working as turbulence promoters. INCONEL-690 is used for the heat exchanger, the main component with the largest wet surface among the ex-vessel regions. Piping is generally made of SS-316L. OSCAR-Fusion allows the simulation of the behavior of these material in ultra-pure water with assigned chemistry condition. In this study, we used levels of hydrogen of the order of 25 $\frac{mL}{kg_{H_2O}}$, which is the usual value found in literature for PWR and able to overwhelm radiolysis. Lithium concentration has been assumed equal to 0.275 ppm in order to have a pH of 7.0 at the working temperatures.

*2.3. Corrosion Modelling*

In OSCAR-Fusion, six media are considered for each control volume [6]:

- Base metal: a structural alloy (in EUROFER-97, stainless steel or Cu alloys) which undergoes general corrosion in contact with the working fluid.
- Inner oxide: it is a passivation dense adherent layer.
- Outer oxide and deposit: a porous outer layer made mainly of agglomerates and crystals due to precipitation processes of ions and particles transported by the cooling fluid.
- Particles: ions may precipitate and agglomerate. They are assumed as spheres with a log-normal diameter distribution.
- Filters and resins: they constitute the purification system. Thanks to the usual high efficiencies, a small fraction of the flow rate is enough to achieve high purification rates. In present simulations factors of the order of 0.5% have been considered.

In general, OSCAR-Fusion solves mass balance equations of the following form [7]:

$$\frac{\delta m_i}{\delta t} + (\dot{m}_{out} - \dot{m}_i) = \sum_{source} J_m - \sum_{sink} J_m \tag{1}$$

where $(\dot{m}_{out} - \dot{m}_i)$ represents the streaming term, i.e., the difference between the incoming flow rate and the outcome for each volume and medium considered. $J_m$ is the source and loss material current determined by the several mechanisms acting on the single medium and element. These involve corrosion, release, erosion, abrasion, dissolution, precipitation, etc.

To model corrosion and release kinetics several approaches are made available in OSCAR-Fusion. The most sophisticated models take into account the corrosion rate variation due to changes in water temperature and pH. In OSCAR-Fusion simulations, the

Moorea's law has been used as the main model for corrosion and release kinetics [6]. This model is based on the definition of two main input parameter: long-term and short-term corrosion/release kinetics. In this work, since experimental corrosion data on the most exotic materials employed in fusion technology, such as EUROFER-97 and CuCrZr alloys, are not available at the operational conditions of EU-DEMO PHTS, SS-316L data have been used. This lead to the a corrosion velocity fo about $2.5 \times 10^{-12} \frac{g}{cm^2 s}$ at the beginning of the scenario. Determining the optimal water chemistry to minimize corrosion is extremely important and new experimental data are required for innovative materials, such as EUROFER-97 and Cu alloys to derive a solid validation base for simulation activities. In this framework, recommended parameters values have been used for EUROFER-97, pure Cu, and CuCrZr alloys. These parameters derive from engineering judgment and professional experience on ITER but validation campaigns oriented to fusion application should be carried out in the near future.

*2.4. Neutron Activation Data*

Neutron activation reaction rates are certainly key parameters in ACP determination. The activation source terms in OSCAR-Fusion which are used in solving the Bateman equation to evaluate isotope inventories must be calculated with extreme accuracy. OSCAR-Fusion, as many other ACP calculation tools, can digest only mono-energetic homogenized activation reaction rates assigned to each under-flux region in the model adopted. Each ACP estimation nuclear fusion application need a preventive Monte Carlo neutron transport simulation to produce activation reaction rates in the several regions of the machine. This preventive calculation of integrated fluxes and reaction rates might not be so straightforward due to the profound difference in the usual geometrical descriptions adopted in Monte Carlo neutron transport and ACP modelling. Furthermore, while in a fission reactor the neutron spectrum can be considered as invariant throughout the whole core, in fusion machines significant spectra gradients take place due to several elastic and inelastic scattering reactions in structural, shielding, breeding, and cooling material of the tokamak periphery.

In this work neutron fluxes in the divertor cassette have been calculated using MCNP (Monte Carlo N-Particle code) using a simplified model adopted for activation calculations.

## 3. Results on DEMO-DIV PHTS

In this section, some results on the EU-DEMO Divertor PHTS are reported to provide preliminary estimation of the radiological hazard which might be encountered in experimental fusion nuclear power plants under current design. The EU-DEMO is a nuclear fusion power plant under design stage which is managed by the Eurofusion Consortium [2]. Other tokamak facilities are under construction, such as ITER [15] (International Thermonuclear Experimental Reactor) and DTT [16] (Divertor Tokamak Test), or in commissioning like JT-60SA [17] (JT60 Super Advanced). All these fusion demonstrators requires ACP estimation during their licensing phase as major contributors to the occupational dose.

As first test-case we picked the EU-DEMO divertor cooling loop in its double circuit configuration. One loop is dedicated for the divertor cassette cooling. The other serves the Plasma Facing Units in Cu and CuCrZr alloys. In the following only the results concerning the cassette PHTS have been reported, which has in-vessel components made of EUROFER-97 only and a flow rate which is of 860 kg/s (much lower than the PFU flow rate which is about of 5319 kg/s).

The EU-DEMO Divertor PHTS have been simplified in OSCAR-Fusion lumping its 48 tokamak sector distributors (one for each Divertor cassette) in four branches. Figure 3 shows a simplified scheme of the model used in OSCAR-Fusion for the divertor cassette PHTS. Roughly 60 regions have been used for the BOP and the in-vessel components. Missing design information, especially those concerning the CVCS layout, have been obviated using fission-derived data.

The plasma pulsed operation mode, which is the key feature of this power plant, has been represented with a simplified scenario constituted by lumped plasma and standby operations. No transitory phases have been considered as trade-off between the computational overhead and the representativeness of the system. Two scenarios have been essentially considered:

1.  The activation scenario: EU-DEMO operation phase is modelled as a continuous operation over 5.2 years (CY) minus 10 days at 30% of the nominal fusion power followed by 10 days pulsed operation with 48 pulses of 4 h at full power and 1 h dwell time in between. The continuous operations is aimed at representing long-lived radionuclide inventories, such as Co-60 and Mn-54. The final pulses are conceived to have a decent representation of short-lived unstable isotopes, such as Mn-56.
2.  The lumped pulsed scenario: it has been conceived to reproduce in a lumped way the actual EU-DEMO scenario in the first operational phase of 1050 days.

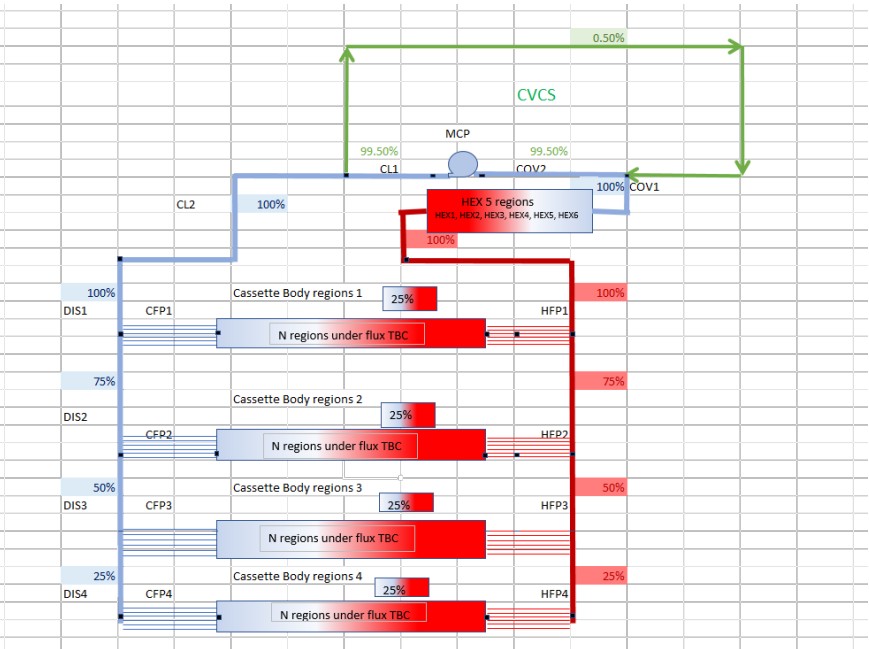

**Figure 3.** EU-DEMO Divertor cassette PHTS simplified scheme of the OSCAR-Fusion model. The 48 branches serving the tokamak cassettes have been simplified in 4 branches. The two loops of the PHTS have been condensed in one.

Figures 4 and 5 show the deposit activities in Bq for the ex-vessel component CL1 (Cold Leg no. 1) and for the heat exchanger at the ex-vessel maintenance times (74, 348, 427, 699, 972, 1051 days). These intervals have been chosen according to the preventive maintenance scheduled for DEMO, where working personnel is called to access the ex-vessel PHTS vaults. Significant radionuclides inventories are already measurable at the first ex-vessel maintenance intervention.

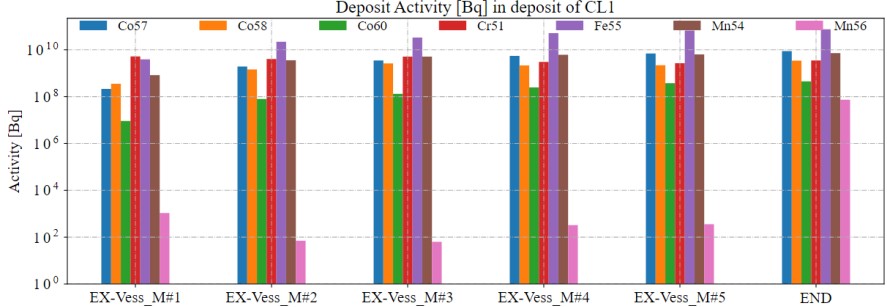

**Figure 4.** EU-DEMO Divertor cassette PHTS Cold Leg no. 1 deposit activity.

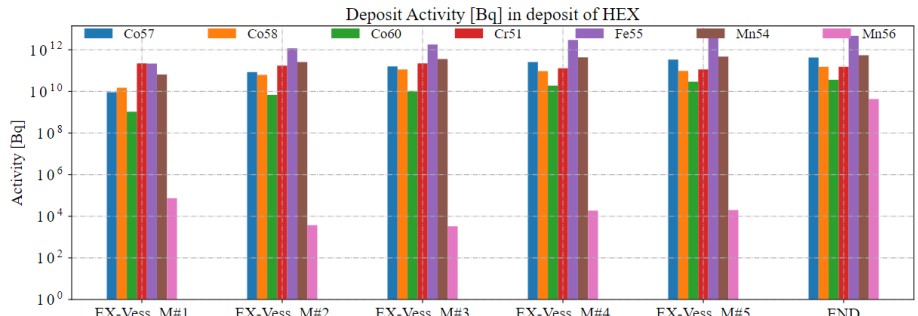

**Figure 5.** EU-DEMO Divertor cassette PHTS Heat Exchanger deposit activity.

A comparison between the evaluated activities with the two scenarios is here considered (see Figures 6 and 7). The activation scenario seems to be more conservative for the deposit activities in the Heat Exchanger and the activities in filters and resins. This kind of result is particularly interesting, showing a quite relevant dependency on the scenario chosen for the calculation. In corrosion problems time constants are different from those characterizing the usual nuclear analyses and an optimal scenario should be considered. Validation procedures for the choice of the most representative scenario in DEMO PHTS analyses will be conceived and performed in future work.

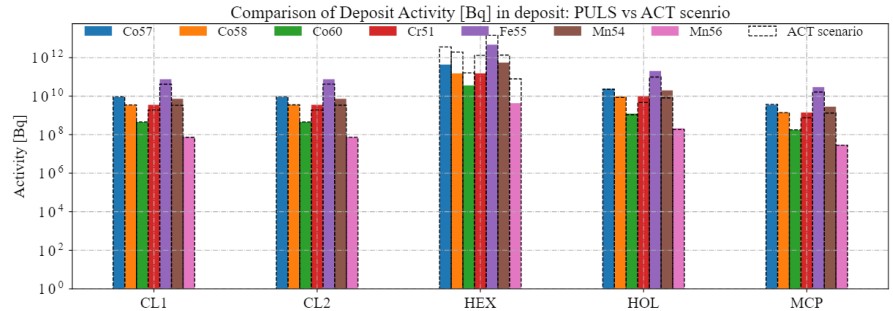

**Figure 6.** EU-DEMO Divertor cassette PHTS ex-vessel deposit activities at the end of the operation scenario. Comparison between the pulsed and the activation scenarios.

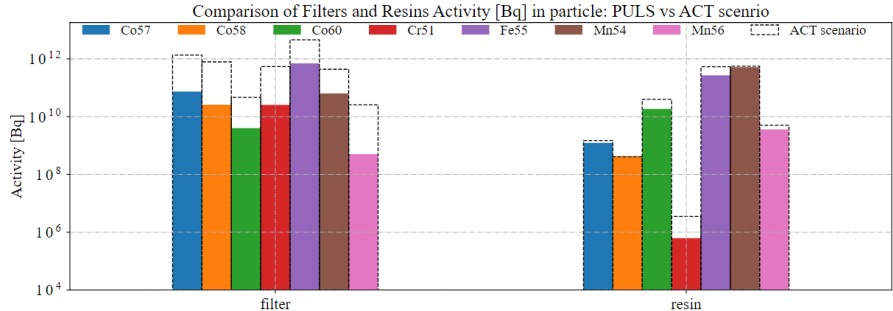

**Figure 7.** EU-DEMO Divertor cassette PHTS filters and resins deposit activities at the end of the operation scenario. Comparison between the pulsed and the activation scenarios.

Ion and particle activities in the cooling fluid (see Figure 8 for particle inventories in suspension) are dominated by Mn-56 which has a quite short half-life (2.5 h). This isotope should be considered together with other important radio-nuclides, such as Co-60 and Mn-54 in accidental scenarios which suppose, as Postulated Initiating Events, loss of coolant accidents.

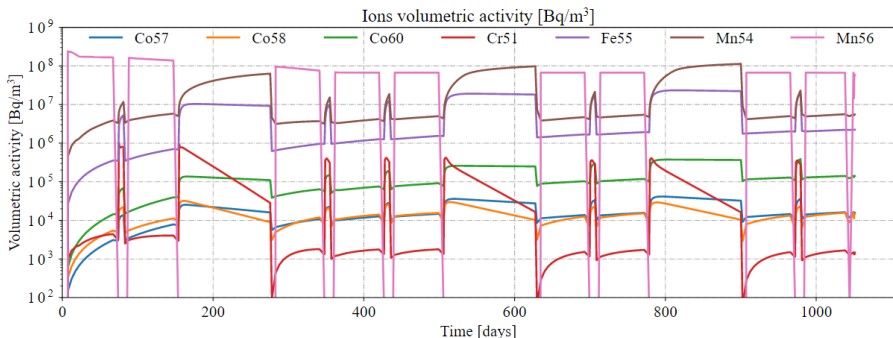

**Figure 8.** EU-DEMO Divertor cassette PHTS activities of ions in solution.

Table 1 shows the numerical values of the deposit activities in Bq during the ex-vessel maintenance for the Heat Exchanger.

**Table 1.** Deposit activities in the different ex-vessel regions at the maintenance scheduled times in Bq.

| Maintenance Period | Co57 | Co58 | Co60 | Cr51 | Fe55 | Mn54 | Mn56 |
|---|---|---|---|---|---|---|---|
| EX-Vess_M#1 | $9.12 \times 10^9$ | $1.5 \times 10^{10}$ | $1.06 \times 10^9$ | $2.24 \times 10^{11}$ | $2.16 \times 10^{11}$ | $6.55 \times 10^{10}$ | 73,618.82 |
| EX-Vess_M#2 | $8.53 \times 10^{10}$ | $6.21 \times 10^{10}$ | $6.84 \times 10^9$ | $1.73 \times 10^{11}$ | $1.19 \times 10^{12}$ | $2.58 \times 10^{11}$ | 3777.598 |
| EX-Vess_M#3 | $1.58 \times 10^{11}$ | $1.14 \times 10^{11}$ | $1.04 \times 10^{10}$ | $2.21 \times 10^{11}$ | $1.81 \times 10^{12}$ | $3.58 \times 10^{11}$ | 3339.743 |
| EX-Vess_M#4 | $2.58 \times 10^{11}$ | $9.40 \times 10^{10}$ | $1.91 \times 10^{10}$ | $1.29 \times 10^{11}$ | $2.95 \times 10^{12}$ | $4.38 \times 10^{11}$ | 18,625.87 |
| EX-Vess_M#5 | $3.38 \times 10^{11}$ | $9.61 \times 10^{10}$ | $2.96 \times 10^{10}$ | $1.13 \times 10^{11}$ | $4.02 \times 10^{12}$ | $4.75 \times 10^{11}$ | 19,758.89 |
| END | $4.30 \times 10^{11}$ | $1.52 \times 10^{11}$ | $3.62 \times 10^{10}$ | $1.52 \times 10^{11}$ | $4.70 \times 10^{12}$ | $5.50 \times 10^{11}$ | $4.33 \times 10^9$ |

As example, if we consider Co-60, we can notice that, at the end of the first operation phase and according to the lumped pulsed scenario, we have $3.0 \times 10^{10}$ Bq. As term of comparison, we could consider IAEA Co-60 D-values [18], corresponding to $3.0 \times 10^{10}$ Bq. Citing [18], "*The D-value is that quantity of radioactive material that, if uncontrolled, could result in the death of an exposed individual or a permanent injury that decreases that person's quality of life. Such health effects are referred to as severe deterministic health effects. Severe deterministic effects usually occur soon after exposure*". D-values are here taken as rough terms of comparison. They refer to concentrated radionuclide inventories in encapsulated or dispersed form. ACPs in tokamak cooling loops, instead, are generally in form of deposits over wet surfaces of thousands of m². Shut down dose rate estimations should be performed to have precise quantification of the radiological hazard, considering that solid deposits are normally attached and confined in the component structures which are normally shielded to minimize occupational exposure. Future work will be devoted to the conversion of ACP inventories to MCNP source terms, aimed at performing photon transport simulations to determine dose rates maps (in μ Sv/h).

## 4. Discussion and Conclusions

In the present manuscript, the general methodology used for ACP estimation in fusion tokamak cooling circuits based on OSCAR-Fusion has been described. ACPs represent a significant radiological hazard which might contribute relevantly to the occupational dose of the workers involved in maintenance operations. The transport of contaminants, in fact, may induce the formation of activated deposits in the ex-vacuum-vessel components, located in vaults where personnel access is allowed. The approach proposed and the simplified model conceived for the divertor cassette PHTS of the EU-DEMO showed significant inventories of ACP with long-lived radionuclides. Co-60 activities are close to what IAEA defines as D-values: critical inventories of radionuclides in different forms which may cause serious damage to humans. ACP results should be, however, accompanied by shut down dose rate calculations which can provide maps in μ Sv/h. Dose maps may be used in fact in collective dose estimations once the maintenance scheduled has

been agreed. Activated deposits and inner oxides layers should be included in photon transport simulations to evaluate dose in the vaults accessible by working personnel. Surface sources are generally implemented in Monte Carlo transport codes, taking into account $\gamma$ and positron emitters (for annihilation X-ray production). Key components are, in fact, normally shielded to limit external exposure of the operators. Workers might, in fact, have direct access to these components even if remote handling operations are foreseen (e.g., maintenance preparation). Filters and resins of the purification systems have to be periodically replaced; pumps, valves, and heat exchangers demand periodical preventive maintenance operations; and, finally, corrective interventions might occur exposing extraordinarily the operators involved.

Different roles are covered by ACP inventories for safety accidental scenarios. In this case, only mobilizable media should be taken into account, such as loosely attached deposits, ions in solution, and particles in suspension. These activated materials should be implemented in severe accident evolution codes, such as MELCOR [19], to provide radiological source terms for the evaluation of the dose to the most exposed individuals. The results obtained in this work have been achieved using provisional corrosion kinetics parameters. The lack of experimental data on corrosion of EUROFER-97 and CuCrZr at the operational conditions of EU-DEMO are imposing assumptions which have to be checked and validated. Recent work [20] showed enhanced corrosion behaviors of EUROFER-97 at pH_300 °C of 7.0 and at pressure and temperatures typical of the Breeding Blanket PHTS (150 bar and 300 °C). The results presented in this manuscript should be revised taking into account new experimental evidence of fusion material corrosion in ultra-pure water. Future work envisages validation procedures of the thermal-hydraulic model and on the corrosion kinetics adopted. ACP inventories will be converted to source terms in MCNP photon transport simulations for dose assessments using the most recent MCNP-OSCAR-Fusion coupling tools developed at ENEA. Sensitivity and parametric analyses aimed at estimating ranges of variation of the final collective dose when changing the OSCAR-Fusion main input parameters will be investigated.

**Author Contributions:** Conceptualization, N.T., L.D.P., G.M. and T.K.; methodology, N.T., L.D.P. and G.M.; software, N.T. and G.M.; validation, N.T., T.K. and S.B.; formal analysis, N.T., T.K. and S.B.; investigation, N.T., T.K., S.B., G.M., C.G. and G.M.C.; resources, N.T. and T.K.; data curation, N.T., T.K., S.B. and G.M.; writing—original draft preparation, N.T.; writing—review and editing, N.T., T.K., C.G. and G.M.C.; visualization, N.T.; supervision, N.T.; project administration, N.T.; funding acquisition, N.T. and T.K. All authors have read and agreed to the published version of the manuscript.

**Funding:** This work has been carried out within the framework of the EUROfusion Consortium, funded by the European Union via the Euratom Research and Training Programme (Grant Agreement No. 101052200—EUROfusion). Views and opinions expressed are however those of the author(s) only and do not necessarily reflect those of the European Union or the European Commission. Neither the European Union nor the European Commission can be held responsible for them.

**Data Availability Statement:** The datasets generated and analyzed in this study are available from the corresponding author on request.

**Acknowledgments:** The authors express their gratitude to Frédéric Dacquait and François Broutin of CEA (Cadarache) for their support in OSCAR-Fusion delivery and utilization.

**Conflicts of Interest:** The authors declare that they have no known competing financial interests or personal relationships that could have appeared to influence the work reported in this paper.

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
