# Peer review of "Activated Corrosion Products Evaluations for Occupational Dose Mitigation in Nuclear Fusion Facilities"

_environments, doi:10.3390/environments9070076_

Round 1

Reviewer 1 Report

The paper discusses an issue that is narrowly specified to the evaluation of activated corrosion products in the nuclear industry. The results are presented very briefly in the conclusion. It will be good to make the conclusions more specific. The paper describes more the issue of radionuclide activities than the actual contribution to the evaluation of corrosion products.  

There are several formal errors in the text:

65 Section ?? 

112 Volume Control System (CVCS) or (VCS)?

132 indentation of a sentence at the edge of a line 

227 invntoreies or inventories?

Of the 17 citations, the authors use 4 of their own citations, where the author is always Luigi Di Pace. I recommend citing from sources of other authors.

I would also recommend to be more specific about the conclusions and what specific contribution the issue under study has, e.g. from a maintenance perspective.

Reviewer 2 Report

This paper presents the assessment on the activated corrosion products from nuclear plant. The research topic is interesting and investigation was relatively reliable. The reviewer thinks it can be considered if the authors could improve this manuscript by the following comments:

(1) The author should highlight the innovation of the global investigation.

(2) More discussion should be involved based on the investigation data.

(3) The presentation was rather tedious, it should be better reorganized in a more concise expression.

Reviewer 3 Report

This paper was well prepared and organized from the point that it showed the possibility of evaluating the activated corrosion products in nuclear fusion facilities. But some part of the manuscript needs to be revised as follows;

Line 59; PACTOLE etc. need the full name.

Line 65; Section ??

Line 74; Please describe the working fluid in detail, not water.

Line 100; Where is 1D regions? What means?

Figure 1 and Figure 2 need the colored ones, if possible.

Line 119; Please show the input data in corrosion modelling. There is no information for the boundary condition in simulation.

Figure 3, Figure 4, Figure 5, Figure 6, Figure 7, and Figure 8 need the colored ones.

Line 121; Do you have the information about the general corrosion of base metals in this system? In Figure 2, please show the location of the alloys used.
